TECHNICAL RELEASE

# PhysiCOOL: A generalized framework for model Calibration and Optimization Of modeLing projects

Inês G. Gonçalves[1,*], David A. Hormuth II[2], Sandhya Prabhakaran[3], Caleb M. Phillips[2] and José Manuel García-Aznar[1]

1 Multiscale in Mechanical and Biological Engineering (M2BE), Aragon Institute of Engineering Research (I3A), University of Zaragoza, Spain
2 Oden Institute for Computational Engineering and Sciences, The University of Texas at Austin, USA
3 Integrated Mathematical Oncology Department, H.Lee Moffitt Cancer Center and Research Institute, USA

## ABSTRACT

*In silico* models of biological systems are usually very complex and rely on a large number of parameters describing physical and biological properties that require validation. As such, parameter space exploration is an essential component of computational model development to fully characterize and validate simulation results. Experimental data may also be used to constrain parameter space (or enable model calibration) to enhance the biological relevance of model parameters. One widely used computational platform in the mathematical biology community is *PhysiCell,* which provides a standardized approach to agent-based models of biological phenomena at different time and spatial scales. Nonetheless, one limitation of *PhysiCell* is the lack of a generalized approach for parameter space exploration and calibration that can be run without high-performance computing access. Here, we present *PhysiCOOL*, an open-source Python library tailored to create standardized calibration and optimization routines for *PhysiCell* models.

**Subjects** Software Engineering, Software and Workflows, Data Integration

**Submitted:** 06 December 2022

**\*** Corresponding author. E-mail: iggoncalves@unizar.es

Preprint submitted at https://doi.org/10.1101/2022.11.17.516671

## STATEMENT OF NEED

Mathematical biology is a field of study that aims to represent biological systems through the language of mathematics: a set of rules that can be used to test hypotheses and make predictions [1]. Several types of mathematical models can be employed to simulate biological systems at varying complexity levels. Agent-based models are among the most popular implementations to develop models that consider the cellular and sub-cellular scales. Currently, multiple computational frameworks are available to facilitate the creation of agent-based models based on previously built templates, making mathematical biology more accessible to researchers from different backgrounds [2]. Among these platforms, *PhysiCell* [3] is an open-source hybrid framework that is able to simulate cells as discrete agents and model the reaction-diffusion dynamics of the substances present in the surrounding microenvironment through a continuous approach. Furthermore, recent add-ons have been developed to introduce new biological processes into the *PhysiCell* ecosystem [4–6].

Despite the recent advances in the development of additional *PhysiCell* plugins, the new modules are mostly centred around model extensions. Nevertheless, model exploration can be as important as model development to validate results and evaluate whether the model predictions about the underlying biological mechanisms are plausible [7]. Furthermore, experimental data could be used to provide biological and/or physical constraints on model parameters to validate whether the model captures the range of expected biological behaviours [8]. Also, optimization routines could be employed to understand which model parameters maximize the similarity between the model results and a target dataset. Subsequently, model developers may consider these optimal solutions to identify which biological mechanisms captured by the computational model may explain the experimental data.

We highlight that previous works have developed parameter exploration routines with *PhysiCell*, namely DAPT (RRID:SCR_021032) and PhysiCell-EMEWS [9, 10], but these were specifically designed for high-performance computing (HPC) and distributed systems. Hence, currently, general *PhysiCell* users without access to such resources, or whose needs do not require them, must develop their own scripts to process simulation results and perform model exploration studies. As well as introducing a barrier to scientific progress depending on the researchers' programming knowledge and computing resources, HPC workflows generally lack the standardization that may enable widespread use in the mathematical biology community [11]. In addition, DAPT and PhysiCell-EMEWS focus on parameter exploration and not optimization, and they require some level of expertise in both Python (RRID:SCR_008394) and PhysiCell.

Taking into account that there is still a need in the *PhysiCell* community for a standardized tool that implements calibration and optimization routines, we present *PhysiCOOL*, a generalized framework for model calibration and optimization of modelling projects written in *PhysiCell. PhysiCOOL* aims to be model agnostic. In other words, models are treated as a black box that can be executed through Python, making this approach suitable for several kinds of biological problems. Moreover, our library includes a built-in multilevel optimization routine for parameter estimation that is constrained by target output (experimental or otherwise). A visual representation of the new functionalities added by *PhysiCOOL* to the *PhysiCell* ecosystem is shown in Figure 1. We also provide two practical examples of how *PhysiCOOL* can be used, showcasing *PhysiCOOL*'s optimization routine at two distinct complexity levels. Furthermore, we show how *PhysiCOOL* black-box models can be used to couple *PhysiCell* with other publicly-available Python libraries for model optimization.

## IMPLEMENTATION

*PhysiCOOL* is a Python library that requires Python version 3.8 or higher. This package was created to work specifically with *PhysiCell* models, and it fully supports *PhysiCell* v1.10.4 or lower (the most recent version at the time of publication). Furthermore, *PhysiCOOL* has been tested extensively and includes unit tests to ensure that its modules are working as expected and that it can be used on different platforms.

## Configuration file parser

As with many computational modelling frameworks, *PhysiCell* models are initialized with values stored in a text-based configuration file, namely an Extensible Markup Language



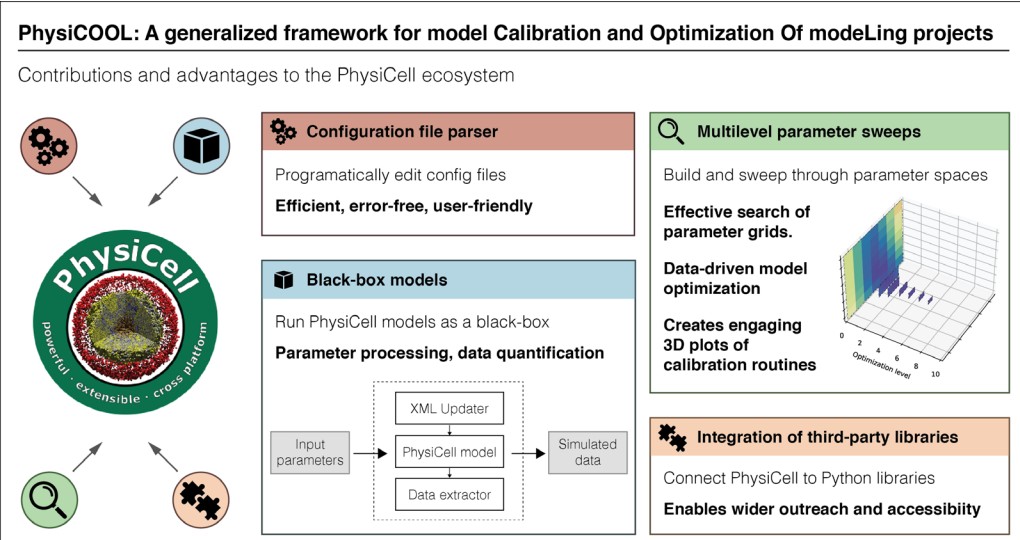

**Figure 1.** *PhysiCOOL*'s contributions and advantages to the *PhysiCell* ecosystem. PhysiCOOL aims to improve the way researchers design and implement their parameter and calibration studies for models written in *PhysiCell*. To this end, *PhysiCOOL* introduces new functionalities, such as a configuration file parser that updates configuration files in an error-free and user-friendly manner. *PhysiCOOL* also enables users to turn models into black-box models, making the optimization pipeline model-agnostic. In addition, it implements a multilevel parameter sweep routine to optimize models using some target data. Lastly, *PhysiCOOL* facilitates the integration of third-party libraries, which makes *PhysiCell* more accessible.

(XML) file [3]. Thus, in parameter sweeps and sensitivity analysis studies, it is necessary to open these files and modify the parameter values to be studied every time a new simulation is run. This process can be done manually, either by editing the XML file directly or using GUI tools such as *xml2jupyter* [12]. However, it becomes unfeasible to repeat this action several times in large-scale studies. Henceforth, it is crucial to automate this process to run optimization and calibration workflows. Although it is possible to create Python scripts that will edit these files automatically with a standard module such as *ElementTree* [13], doing so requires users to identify the values to be updated with long strings that reflect the structure of the XML file, as shown in the code snippet below.

```python
from xml.etree import ElementTree

# Read cell data
file_path = "config/PhysiCell_settings.xml"
tree = ElementTree.parse(file_path)

# Define where to find the motility parameters
stem = "cell_definitions/cell_definition[@name='default']/phenotype/motility"
# Define the name and value of the parameter to be updated
key = "migration_bias"
value = 0.9
# Update the migration_bias value (no validation)
tree.find(f"{stem}/{key}").text = str(value)
tree.write(file_path)
```

**Table 1.** Data classes implemented in PhysiCOOL.

| Class | Variable name | Description | Type | Constraints |
|---|---|---|---|---|
| Domain | x_min/y_min/z_min | Lower bound of the domain | float | - |
| | x_max/y_max/z_max | Upper bound of the domain | float | - |
| | dx/dy/dt | Domain voxel length | float | >0 |
| Overall | max_time | Total simulation time | float | >0 |
| | dt_diffusion/dt_mechancis/dt_phenotype | Time between simulation events | float | >0 |
| Substance | name | Substance name | string | - |
| | diffusion_coefficient | Diffusion coefficient | float | >0 |
| | decay_rate | Decay rate | float | >0 |
| | initial_condition | Initial conditions for the entire domain | float | >0 |
| | dirichlet_boundary_condition | Dirichlet boundary conditions | float | >0 |
| Cell cycle | phase_durations | Phase duration list | list | >0 |
| | phase_transition_rates | Phase transition rates list | list | >0 |
| Cell death | phase_durations | Phase duration list | list | >0 |
| | phase_transition_rates | Phase transition rates list | list | >0 |
| Cell volume | total | Total cell volume | float | >0 |
| | fluid_fraction | Cell fluid fraction | float | [0–1] |
| | nuclear | Cell nuclear volume | float | >0 |
| | fluid_change_rate | Fluid change rate | float | >0 |
| | cytoplasmic_biomass_change_rate | Cytoplasmic biomass change rate | float | >0 |
| | nuclear_biomass_change_rate | Nuclear biomass change rate | float | >0 |
| | calcified_fraction | Calcified fraction | float | [0–1] |
| | calcification_rate | Calcification rate | float | >0 |
| | relative_rupture_volume | Relative rupture volume | float | >0 |
| Cell mechanics | cell_cell_adhesion_strength | Adhesion strength | float | >0 |
| | cell_cell_repulsion_strength | Repulsion strength | float | >0 |
| | relative_maximum_adhesion_distance | Maximum adhesion distance | float | >0 |
| Cell motility | speed | Cell speed | float | >0 |
| | persistence_time | Mean persistence time | float | >0 |
| | migration_bias | Migration bias | float | [0–1] |

Here, we aimed to develop a Python class that enables users to read the data from these configuration files more efficiently, making this process less prone to errors. We implemented a *ConfigurationFileParser* class that reads the data from the configuration file into custom Python objects that follow the expected structure and data requirements defined in the XML file. A more detailed description of this implementation is presented in Table 1. Variable types and numerical constraints are validated when new instances of these data classes are created and when their values are updated. To achieve this, we implemented our classes using *Pydantic* [14], which improves data validation in Python. The task described in the code snippet presented previously can be implemented in a more user-friendly way with *PhysiCOOL*, as shown below:

```python
from physicool.config import ConfigFileParser

# Read cell data into custom Python objects
file_path = "config/PhysiCell_settings.xml"
parser = ConfigFileParser(file_path)
cell_data = parser.read_cell_data(name="default")

# Update the migration_bias value (values will be validated before writing)
cell_data.motility.migration_bias = 0.9
parser.write_cell_params(cell_data)
```

## Black-box models

In complex and large computational models, it may be challenging or even impossible to estimate the model outputs analytically. Consequently, it is common to conduct calibration



and optimization studies by running several simulations and performing sensitivity analysis studies to identify how model outputs change in response to different input parameter values. This process is recognized as simulation-based optimization or black-box optimization [15]. _PhysiCell_ models are written in C++ and have to be compiled to produce an executable file that can be run to produce simulation results. In order to test and characterize the response of these models, it is generally necessary to conduct three tasks:

 (i) Update the PhysiCell configuration file with input parameter values;
 (ii) Run the PhysiCell model;
(iii) Read the model outputs and compute a desired output metric.

 These tasks can be performed manually. Nonetheless, it is not feasible or productive to do so in large computational studies, specifically when trying to characterize the model response to a large number of input parameter values that can be inside a wide range and require multiple simulation runs. Hence, PhysiCOOL allows users to create black-box models using the _PhysiCellBlackBox_ class and automatically perform the aforementioned tasks through Python.

 These black-box models are modular in that the users can select what functions to use to update the configuration file (i) and process the results (iii). For instance, users can decide to change the cells' motility parameters and evaluate the effect on the distance travelled by cells over time. Alternatively, the cell cycling rates could be varied to analyze the evolution of the number of cells. Furthermore, (i) and (iii) do not have to be defined in the black-box model. In fact, users can also create black-box models composed only of the PhysiCell executable and use our approach to run multiple simulation replicates.

 PhysiCOOL offers some built-in data quantification methods that can be used to extract and process data in step (iii). For example, functions are provided to obtain the final number of cells in a simulation, the final cell coordinates, and the concentration of a given substance over the simulation domain. Furthermore, these methods can be employed by users to process simulation results and generate 2D and 3D plots of the cells and the microenvironment.

## Multilevel parameter sweeps

Parameter optimization studies require the definition of a search space, which defines the range of the parameter values that will be studied. There are multiple approaches to defining this space and how to explore it. For example, random search algorithms can be employed to randomly sample points within a defined bounded parameter space. Alternatively, a grid search, while a more computationally expensive option, systematically samples every point within a defined parameter grid space providing a more comprehensive overview of the model's response than that offered by a random search. Grid-based approaches have advantages for stochastic frameworks such as PhysiCell, as gradient-based approaches may struggle to accurately calculate the gradient and change the parameter set to minimize the error between the model and the target data.

 PhysiCOOL implements a multilevel parameter sweep class (_MultiLevelSweep_) that is aimed at identifying the parameters that best fit a target dataset through a grid search. In this example, the parameter sweep considers two PhysiCell parameters for which the user should provide initial values. At each level, _MultiLevelSweep_ creates a search grid based on

**Table 2.** Parameter values used in the multilevel optimization examples.

| Example | Initial point | Points per direction | Percentage per direction | Levels | Estimated point | Target point |
|---|---|---|---|---|---|---|
| Logistic growth | (0.15, 1000.0) | 8 | 50% | 7 | (0.10, 994.7) | (0.10, 1000.0) |
| Chemotaxis | (2.5, 0.7) | 5 | 30% | 4 | (1.7, 0.8) | (2.0, 0.9) |

these two values, the number of points per direction and the percentage per direction. These values should be configured by the user and optimized for a given problem. Furthermore, the number of levels and grid spacing parameters are related to the precision and sensitivity of each model parameter. That is, for less sensitivity or less precise models, a single-level coarse grid search may suffice. However, for parameters that require a high level of precision and significantly affect the model outcomes, multiple levels may be beneficial.

The results for each simulation are compared to the target data, and the error between both datasets is computed and stored. At the end of the level, the parameters that provide the minimum error value are selected as the centre of the parameter exploration grid for the next level, and the parameter bounds are updated accordingly.

## EXAMPLES

## Simple model of logistic growth

The first example was implemented to calibrate two parameters of a simple model of logistic growth based on target data that defines a generated growth curve. Therefore, it serves as an introduction to this *PhysiCOOL* feature, as users are able to fully understand the behaviour of this simple model. It must be remarked that this model was not implemented in *PhysiCell*. We modelled the number of agents in a population, *N*, over a period of time *t* through a logistic function given by Equation (1):

$$N(t) = \frac{KN_0}{N_0 + (K - N_0)\exp(-rt)} \tag{1}$$

where $K$ represents the carrying capacity, i.e., the maximum population size, $N_0$ represents the number of initial agents and $r$ is the proliferation rate. In this study, we fixed the initial number of agents and evaluated how the carrying capacity and the proliferation rate regulated the growth curve of a population. An example of two growth curves obtained for different model parameters is shown in Figure 2(a).

We generated some target data using this model ($K = 1000$, $r = 0.1$) and, subsequently, we used *PhysiCOOL*'s multilevel sweep algorithm to evaluate if we could estimate these model parameters based on their resulting growth curve. To do so, we first created a search grid based on a set of user-defined values: an initial estimate for both parameters, the number of points to search in each direction of the search grid, the percentage to vary in each direction and the number of levels to search. These values can be found in Table 2.

Figure 2(b) shows the error between the target and simulated datasets for every cell of the parameter space after one level of the multilevel search. At this point, a new point estimate was calculated based on the parameter values that minimized the error between the two datasets. Likewise, the parameter space was adjusted to the area of interest and the process was repeated in the new parameter grid. This process was repeated for each level of the search and the results are shown in Figure 2(c).



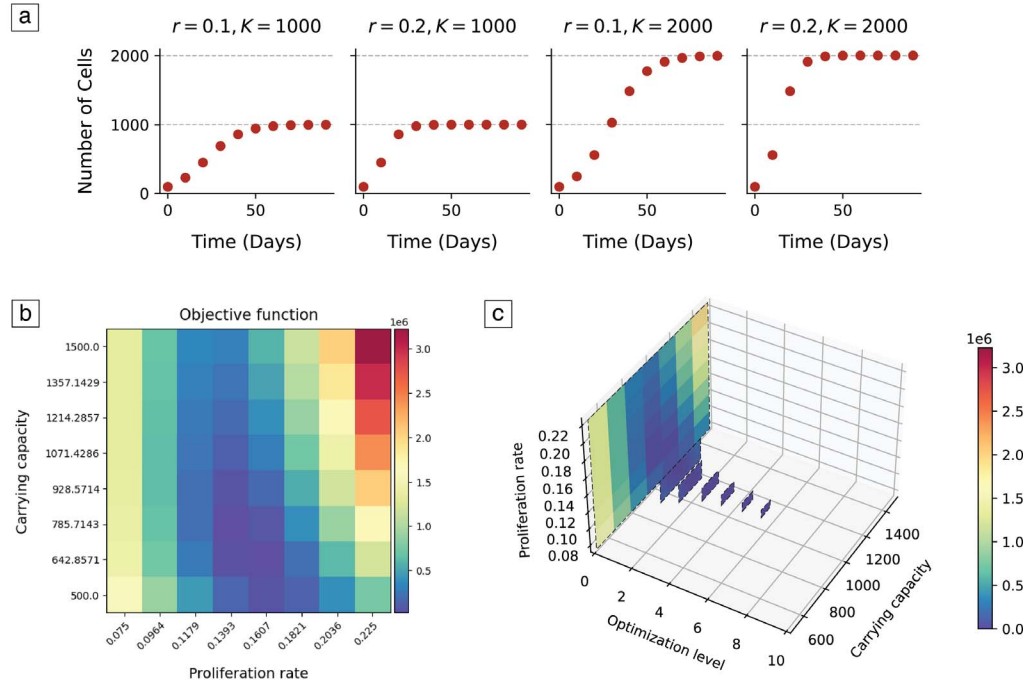

**Figure 2.** Model and optimization results for the logistic growth example. (a) Growth curves obtained for different parameter sets (carrying capacity, *K*, and proliferation rate, *r*). (b) Optimization results after the completion of the first level of the multilevel optimization algorithm. The heatmap shows the difference, as given by the summed squared error, between the target data and the data produced by each cell's input parameters. (c) Optimization results after seven levels of the multilevel optimization algorithm. Results converged to the parameters that originated the target data.

## PhysiCell chemotaxis model

The second example can be classified as a more complex problem since it was developed to calibrate a chemotaxis model written in *PhysiCell*. In this modelling framework, the cells' chemotactic response, i.e., the ability to migrate along a substance gradient, is dictated by a bias value defined between 0 and 1 [3]. When cells have a migration bias of 0, they move in a random walk. Conversely, if the value is set to 1, cells follow the substance gradient in a deterministic manner. Therefore, we developed a model to estimate the cells' speed and migration bias in response to an oxygen gradient based on their travelled distances.

We implemented a 2D simulation with an oxygen source on one of the domain walls, as defined by the model's boundary conditions, and a group of cells placed on the opposite wall, as shown in Figure 3(a). We expected that the cells' final position would be modulated by the cells' sensitivity to the oxygen chemotactic gradient. On the one hand, if a cell population had low sensitivity and, thus, moved randomly, they would likely remain close to their initial position as they would move around without following any specific direction. On the other hand, cells that followed oxygen would move towards the opposite wall, as seen in Figure 3(b).

We generated some target data by running a simulation with a migration bias of 0.9 and a speed value of 2.0 µm/min and storing the final y coordinates of the cells. Subsequently, we ran our multilevel sweep pipeline to evaluate whether we could estimate the parameter values that originated this data with a set of initial points different from the target parameter values. The results of this study are shown in Figure 3(c).



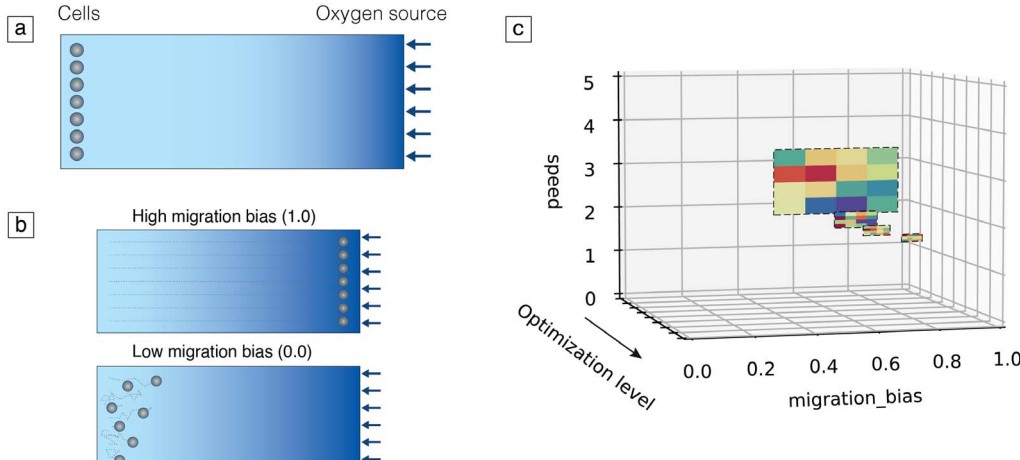

**Figure 3.** Model and optimization results for the chemotaxis example. (a) Initial model configuration design. Cells (represented as grey circles) were placed close to a domain wall and an oxygen source (represented by the blue arrows) was simulated on the opposite wall, creating a chemotactic gradient that cells could follow. This gradient is illustrated by the colour gradient shown in the figure. (b) Expected model results for cells with different migration bias values. High migration bias populations were expected to migrate in a deterministic manner and follow the oxygen gradient, crossing the domain and arriving at the opposite wall. Cell trajectories are shown as grey dashed lines. On the other hand, cells with low migration bias were expected to move randomly and, thus, present low net displacement values. (c) Optimization results after four levels of the multilevel optimization algorithm. Results converged to the parameters that originated the target data. The colourmap was updated for each level, describing the minimum and maximum error values at the current level.

## Connecting to third-party libraries

*PhysiCOOL* makes it possible for users to turn their *PhysiCell* models into black-box models that receive some input parameters and return an output metric. Hence, it is straightforward to couple them with third-party Python libraries that accept this kind of model. For example, *psweep* [16] is a Python library developed to run parameter studies and save the input parameter values and the returned output metrics into a database. Users must define a set of parameters and, for each of the defined values, *psweep* will (i) run a given user-defined function that takes these parameters as input and (ii) save the input and output values returned by this function into the database. Therefore, a PhysiCOOL black-box model could seamlessly be integrated into step (i).

In addition, more sophisticated libraries could be considered to perform advanced optimization studies, such as Approximate Bayesian Computation (ABC) and Bayesian Optimization for Likelihood-Free Inference (BOLFI), to sample parameter spaces in a more efficient manner [17–19]. Henceforth, although *PhysiCOOL* offers built-in optimization routines, it can be used in a modular way to take advantage of other libraries that may be more appropriate to a certain study or type of research, without the need to implement these optimization algorithms from scratch.

## FUTURE DIRECTIONS

At its current state of development, we believe that *PhysiCOOL* already improves *PhysiCell*'s accessibility as it provides an intuitive interface to run studies in Python, which is more popular among biology researchers than C++, in which *PhysiCell* was originally written. Additionally, this standardized approach provides a straightforward workflow for integrating target data (defined from simulations or biological observations) to constrain

the parameter space for agent-based models. In the future, new features can be added to *PhysiCOOL*, such as the ability to generate non-linear parameter spaces, stopping criteria based on iteration or tolerance for the multilevel sweep and employing alternative optimization algorithms. Although future iterations of this library may include different optimization approaches, its modular design assures that advanced users are still able to build pipelines that suit their needs.

## AVAILABILITY OF SOURCE CODE AND REQUIREMENTS

- Project name: PhysiCOOL
- Project home page: https://github.com/IGGoncalves/PhysiCOOL
- Operating system(s): Platform independent
- Programming language: Python
- Other requirements: Python >=3.8, PhysiCell <=1.10.4
- License: BSD 3-clause license
- RRID:SCR_023305.

All the examples presented here (logistic growth, chemotaxis and connecting to third-party libraries) can be run online through interactive Jupyter Notebooks found in https://gitpod.io/#https://github.com/IGGoncalves/PhysiCOOL (see the *examples* folder for more information). Documentation and further information on *PhysiCOOL* can be found on ReadTheDocs (https://physicool.readthedocs.io/en/latest). Users may seek help, report issues and suggest improvements through the GitHub issues page of the *PhysiCOOL* repository.

## DATA AVAILABILITY

Snapshots of the code and data are available in the GigaDB repository [20].

## DECLARATIONS

### List of Abbreviations

HPC: high-performance computing.

### Ethical approval

Not applicable.

### Competing Interests

The authors declare that they have no competing interests.

### Funding

This work was supported as part of the 2021 PhysiCell Hackathon (administrative supplement to Multiscale systems biology modeling to exploit tumor-stromal metabolic crosstalk in colorectal cancer, grant no 1U01CA232137). IGG and JMG-A were supported as part of projects that have received funding from the European Research Council (ERC) under the European Union's Horizon 2020 research and innovation programme (grant agreement no 101018587) and the project PRIMAGE (PRedictive In-silico Multiscale Analytics to support cancer personalized diaGnosis and prognosis, empowered by imaging biomarkers), a Horizon 2020 | RIA project (Topic SC1-DTH-07-2018), grant agreement no 826494.

## Authors' contributions

- Conceptualization: IGG, DAHII, CMP, SP
- Software: IGG, DAHII, CMP
- Validation: IGG, DAHII, SP
- Writing - original draft: IGG
- Writing - review & editing: IGG, DAHII, SP, JMG-A
- Funding acquisition: JMG-A.

## Acknowledgements

The authors would like to acknowledge Paul Macklin, Elmar Bucher and the PhysiCell team for the support and advice offered during the design process of this application.

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
