## [Reviewer Report]

Comments on revised manuscriptThe authors addressed all my concerns and I have no further reservations in recommending this manuscript for publication.

---

## [Reviewer Report]

Reviewer name and names of any other individual's who aided in reviewerCicely Krystyna MacnamaraDo you understand and agree to our policy of having open and named reviews, and having your review included with the published manuscript. (If no, please inform the editor that you cannot review this manuscript.)YesIs the language of sufficient quality?YesPlease add additional comments on language quality to clarify if neededIs there a clear statement of need explaining what problems the software is designed to solve and who the target audience is? YesAdditional CommentsThis is set out extremely clearly in the paper.Is the source code available, and has an appropriate Open Source Initiative license <a href="https://opensource.org/licenses" target="_blank">(https://opensource.org/licenses)</a> been assigned to the code?YesAdditional CommentsThe code is freely available on Git.As Open Source Software are there guidelines on how to contribute, report issues or seek support on the code?NoAdditional CommentsThis is not explicitly stated in the paper although it is possible to seek support via GitHub.Is the code executable?Unable to testAdditional CommentsI was able to run my own executable files and can confirm that these ran well and quickly with the code (e.g. running a batch of simulations). Although everything appears in order I had some difficulties using the provided executables and so was unable to test the examples provided (e.g. growth)Is installation/deployment sufficiently outlined in the paper and documentation, and does it proceed as outlined?YesAdditional CommentsInstallation of PhysiCool is quick and easy.Is the documentation provided clear and user friendly?YesAdditional CommentsPlease see comment below but otherwise the documentation is clear and well-presented.Is there enough clear information in the documentation to install, run and test this tool, including information on where to seek help if required?NoAdditional CommentsConsidering the authors wish this tool to be used by the non-expert user I felt there could be more details given. It is simple if you already work with python, jupyter notebooks and PhysiCell but in order to make this tool valuable to a general audience a more clear step-by-step guide could be provided. Is there a clearly-stated list of dependencies, and is the core functionality of the software documented to a satisfactory level?YesAdditional CommentsHave any claims of performance been sufficiently tested and compared to other commonly-used packages? Not applicableAdditional CommentsNo claims of performance included in paper.Is test data available, either included with the submission or openly available via cited third party sources (e.g. accession numbers, data DOIs)?YesAdditional CommentsAre there (ideally real world) examples demonstrating use of the software? YesAdditional CommentsIs automated testing used or are there manual steps described so that the functionality of the software can be verified?YesAdditional CommentsTo some extent (see above).Any Additional Overall Comments to the AuthorThe manuscript entitled " PhysiCOOL: A generalized framework for model Calibration and Optimization Of modeLing projects" is succinctly written; its purpose is clear and the software created simple yet effective. I think improvements could be made to the documentation allowing a non-expert user to make use of this valuable tool. I also have a few minor comments below. Otherwise I am happy to recommend the publication of this paper.  Minor comments:  (1) Could the authors clarify in the paper (where it says PhysiCool has partial support for PhysiCell v1.10.3 and higher) whether it is the author's intention to keep this tool up to date with newer releases of PhysiCell?  (2) For the multilevel parameter sweep the authors suggest that the number of levels and grid parameters can be defined by the user. Do the authors have any suggestions on picking the appropriate number of levels, for example, or could future development include some form of dynamic choice for number of levels e.g. stop when a certain degree of accuracy is found?RecommendationMinor Revisions

---

## [Reviewer Report]

Reviewer name and names of any other individual's who aided in reviewerDaniel BergmanDo you understand and agree to our policy of having open and named reviews, and having your review included with the published manuscript. (If no, please inform the editor that you cannot review this manuscript.)YesIs the language of sufficient quality?YesPlease add additional comments on language quality to clarify if neededIs there a clear statement of need explaining what problems the software is designed to solve and who the target audience is? YesAdditional CommentsIs the source code available, and has an appropriate Open Source Initiative license <a href="https://opensource.org/licenses" target="_blank">(https://opensource.org/licenses)</a> been assigned to the code?YesAdditional CommentsAs Open Source Software are there guidelines on how to contribute, report issues or seek support on the code?NoAdditional CommentsIs the code executable?NoAdditional CommentsI attempted to run the motility example by performing the following steps on an M1 Mac: 1) download PhysiCOOL from Github. 2) Open black_box.ipynb 3) Successfully run code blocks up to (not including) "Plotting the results" 4) First block in "Plotting the results" returns an error; it seems there is no data for it to use.   I have been able to confirm that a "temp" folder is created upon calling black_box.run, but it is not being populated with output files. Hopefully, I am only having this issue as an inexperienced python user, but I'm still unsure how to fix this myself.Is installation/deployment sufficiently outlined in the paper and documentation, and does it proceed as outlined?YesAdditional CommentsThe documentation is very clear on what to do. See above for the issue I ran across in running the examples.Is the documentation provided clear and user friendly?YesAdditional CommentsHaving the examples in the documentation would be helpful. Currently, they are only on Github.Is there enough clear information in the documentation to install, run and test this tool, including information on where to seek help if required?YesAdditional CommentsNo information is provided on where to seek help.Is there a clearly-stated list of dependencies, and is the core functionality of the software documented to a satisfactory level?YesAdditional CommentsHave any claims of performance been sufficiently tested and compared to other commonly-used packages? Not applicableAdditional CommentsIs test data available, either included with the submission or openly available via cited third party sources (e.g. accession numbers, data DOIs)?YesAdditional CommentsAre there (ideally real world) examples demonstrating use of the software? YesAdditional CommentsIs automated testing used or are there manual steps described so that the functionality of the software can be verified?YesAdditional CommentsAny Additional Overall Comments to the AuthorThis is a very nice addition to the PhysiCell ecosystem. Methods for parameterizing agent-based models is critical, and the ability to do so without expensive computing resources, i.e. HPC, will aid many researchers.   Comments: 1) "Furthermore, experimental data could..., they can be used..." this feels like a run-on sentence. It is unclear who/what "they" is.  2) "bespoke HPC workflows..." Is this referencing DAPT and the PhysiCell-EMEWS workflow? If so, how does PhysiCOOL differ from these?  3) Is PhysiCOOL defining this multilevel sweep approach to parameter estimation? Or is this already established? If the former, please emphasize. If the latter, are there citations? 4) Please emphasize that the "Simple model of logistic growth" is not done with PhysiCell. 5) I needed Python version < 3.11.0 to install physicool  Major revisions: 1) Please check on the issue I had with the motility example and it not generating output files.  Minor revisions: 1) "As for many several computational modelling frameworks..." consider rewording. I would suggest "As with many computational modeling frameworks" 2) "...namely **an** Extensible..."  3) "...can be employed to randomly **sample** points within..." 4) Please change notation in Table 2 so that the "* point" columns report the values as coordinates ( , ) rather than like intervals [ , ].RecommendationMajor Revisions